# An Intelligent Fire-Protection Coating Based on Ammonium Polyphosphate/Epoxy Composites and Laser-Induced Graphene

**DOI:** 10.3390/polym13060984

**Published:** 2021-03-23

**Authors:** Weiwei Yang, Ying Liu, Jie Wei, Xueli Li, Nianhua Li, Jiping Liu

**Affiliations:** School of Materials Science and Engineering, Beijing Institute of Technology, Beijing 100081, China; yangweiwei0811@163.com (W.Y.); yingliu@bit.edu.cn (Y.L.); jie_weiwj@163.com (J.W.); 15733185216@163.com (X.L.); 17888818050@163.com (N.L.)

**Keywords:** fire protection, ammonium polyphosphate and epoxy composite, laser-induced graphene, temperature sensor, shape memory effect

## Abstract

Fire-protection coatings with a self-monitoring ability play a critical role in safety and security. An intelligent fire-protection coating can protect humans from personal and property damage. In this work, we report the fabrication of a low-cost and facile intelligent fire coating based on a composite of ammonium polyphosphate and epoxy (APP/EP). The composite was processed using laser scribing, which led to a laser-induced graphene (LIG) layer on the APP/EP surface via a photothermal effect. The C–O, C=O, P–O, and N−C bonds in the flame-retardant APP/EP composite were broken during the laser scribing, while the remaining carbon atoms recombined to generate the graphene layer. A proof-of-concept was achieved by demonstrating the use of LIG in supercapacitors, as a temperature sensor, and as a hazard detection device based on the shape memory effect of the APP/EP composite. The intelligent flame protection coating had a high flame retardancy, which increased the time to ignition (TTI) from 21 s to 57 s, and the limiting oxygen index (LOI) value increased to 37%. The total amount of heat and smoke released during combustion was effectively suppressed by ≈ 71.1% and ≈ 74.1%, respectively. The maximum mass-specific supercapacitance could reach 245.6 F·g^−1^. The additional LIG layer enables applications of the device as a LIG-APP/EP temperature sensor and allows for monitoring of the deformation according to its shape memory effect. The direct laser scribing of graphene from APP/EP in an air atmosphere provides a convenient and practical approach for the fabrication of flame-retardant electronics.

## 1. Introduction

In numerous technological applications and devices, unwanted ignition poses a serious risk of causing harm to human lives and damaging the respective application or device. To prevent and tackle fire hazards, considerable research has been carried out on flame protection, fire extinguishers, and fire surveying or warning technologies [1,2,3,4,5,6,7,8]. For flame-retardant technologies, polymer materials have commonly been applied with the purpose of delaying or avoiding the occurrence of fire [9,10]. For example, flame-retardant epoxy resin is widely used in electronic devices, which prevents electrical appliances from burning or at least reduces the burning speed [11,12]. However, ideal fire-protection coatings should not only be able to protect the material itself from ignition but also have additional smart properties, such as the ability to monitor the material temperature and detect potential hazards. However, it is still a big challenge to realize the functional integration of intelligent self-monitoring fire protection in a single device with multiple functionalities, namely, fire protection, temperature detection, and hazard monitoring. This becomes even more challenging if the respective device requires reducing the size of the fire coating and integrating it into microelectronics.

High-performance polymer materials, such as a flame-retardant epoxy resin (FR-EP), are an indispensable fire-protection coating due to their excellent dimensional stability and high mechanical strength [13,14]. For the purpose of integrating other functionalities into FR-EP materials, a viable strategy is to add additional functional materials into the FR-EPs to form composites. The traditional process consists of spraying, embedding, or filling, where such functional coatings or fillers can be carbon nanotubes, graphene, Ag, and carbon black, which can be deposited directly onto the surface of the polymer [15,16,17,18]. In order to integrate a functionalized temperature sensor via conventional means, multistep routes and high-cost deposition materials are required [19]. Consequently, it is still challenging to prepare smart fire-protection coatings, especially those with straightforward and facile fabrication and custom design capabilities.

To circumvent such tedious multistep composite fabrication routes, graphene nanomaterials may be considered a viable alternative. Recently, graphene nanomaterials have attracted great attention in many areas of industry and academia due to their high conductivity, light weight, marked barrier effect, and excellent stability, which makes them a promising candidate material for the preparation of smart electronic coatings [20,21]. Traditional graphene-based electronic components are fabricated by spraying, embedding, or filling the prepared graphene onto or into the substrate [22]. Tang et al. [23] describe the creation of multifunctional coatings through assembling graphene oxide (GO)/silicone onto flammable substrate materials, which can detect a change in temperature and protect the flammable materials from combustion while the fire occurs. However, the fabrication of a GO/silicone coating needs different reagents, which may lead to a waste of reagents and waste liquid treatment, and requires a complicated preparation process. To simplify the preparation process and reduce the cost, Lin et al. [24] developed a novel, facile, and low-cost approach to generating 3D porous graphene through laser scribing on polyimide. More recently, several research groups reported the integration of graphene onto the surface of polymers, such as those of the polysulfone class and PEEK (poly-ether-ether-ketone) polymers [25,26,27]. The laser-scribing approach that is used to generate graphene is facile, efficient, and has a low cost. Due to the similar polymer structures used in this study to create FR-EP with the abovementioned precursors, we concluded that laser-scribing technology may be an interesting tool to be used with FR-EP, which may facilitate the preparation of graphene-based FR-EP materials that can be integrated into electronic devices as a fire-protection coating composite.

In this work, we propose a facile strategy for preparing a fire-protection coating by integrating direct laser-induced graphene (LIG) into ammonium polyphosphate epoxy resin (APP/EP) composites. The formation of graphene in APP/EP can be attributed to the aromatic structure of the EP resin and the photothermal effect generated by the laser radiation. The resulting LIG-APP/EP device had porous graphene on the surface, which could be obtained in an ambient atmosphere in an optimized experimental process that included multiple laser-scribing process steps (laser power: 5 W, laser-scribing speed: 10 mm·s^−1^). This coating could play both a flame-retardant role and detect potential hazards through the high conductivity of the LIG and the shape memory effect of the APP/EP. The LIG-APP/EP composite exhibited high electrical conductivity (9 Ω, 1 cm × 1 cm), as required for applications in the field of electronics. The potential for application of the LIG-APP/EP composite was demonstrated in N- and P-doped supercapacitor electrodes and temperature sensors, and the hazard detection functionality was shown to be based on the shape memory effect of the APP/EP. The LIG-APP/EP may be regarded as an intelligent coating that has tremendous potential for applications in various fields.

## 2. Materials and Methods

### 2.1. Preparation of LIG via Laser Scribing on APP/EP

Epoxy resin (E44) and m-phenylene diamine (m-PDA) were purchased from Sinopec Baling Petrochemical Company (Yueyang, China) and ammonium polyphosphate (APP: *n* > 2000) was procured from Qingyuan Pusaifurop Chemical Co., LTD (Qingyuan, China). The three materials were mixed in a beaker together until they were uniformly distributed using a hot plate with a magnetic stirrer at a high speed (1000 r/min). The contents of the added APP were 5%, 10%, 15%, and 20%. The mixture of E44, m-PDA, and APP was poured into a mold and then cured at 80 °C for 2 h and at 120 °C for another 2 h, followed by slow cooling to room temperature, and subsequently, the splines were removed. The size of the prepared splines could be controlled using the size of the mold. In this work, three sizes of splines were prepared for testing, which were 130 mm × 6.5 mm × 3 mm, 130 mm × 13 mm × 3 mm, and 100 mm × 100 mm× 1.2 mm. The spline with the size of 130 mm × 13 mm × 3 mm was divided into several small parts for the LIG characterization tests and applied as a temperature sensor in ambient and water conditions.

Laser scribing on the APP/EP composites was performed using 50 W, 10.6 μm CO_2_ laser equipment (Ketai 4060 purchased from Shangdong Kaitai Laser Equipment Co., Ltd., Liaocheng, China). The laser power and laser-scribing speed were set by the computer connected to the CO_2_ laser equipment. After repeatedly verifying the laser parameters for preparing the LIG, we finally determined that the laser parameters for preparing the LIG in this article were as follows: the laser power was adjusted to ≈5 W and the laser-scribing speed was set to ≈10 mm·s^−1^ for all specimens. The laser-scribing process and the temperature sensor tests were carried out in ambient air conditions. In the following statements in this article, the prepared graphene is called LIG, the sensor device is called LIG-APP/EP, and the flame-retardant coating spline is called LIG/APP/EP.

### 2.2. Characterization of the LIG Formed on the APP/EP

The surface morphology and structure of the material were analyzed using scanning electron microscopy (SEM; JSM-IT300, Beijing, China) and transmission electron microscopy (TEM; FEI Tecnai G2 F30, Beijing, China). Raman spectra were collected using a Micro Raman imaging spectrometer (DXRxi, Shanghai, China) with a wavelength of ≈532 nm. X-ray diffraction (XRD) was carried out (D8 advance, Brook, Germany) and X-ray photoelectron spectroscopy (XPS) spectra were measured (Thermo Kalpha, Shanghai, China). Thermogravimetric analysis thermograms were recorded (Q50 TGA Instruments, Minneapolis City, Minnesota, USA) within the temperature range of 100–900 °C at 10 °C/min under an argon atmosphere. The conductivity of the LIG was measured using a standard resistance tester (YD 2511A, Beijing, China). The LIG was scratched off from the APP/EP substrate for further XRD, XPS, and thermogravimetric analyses. Other general characterizations were directly performed on the LIG surface.

The limiting oxygen index (LOI) of the LIG/APP/EP as a flame retardant coating was tested using JF-3 Oxygen Index Equipment, which was purchased from Nanjing Jiangning Analytical Instrument Co., Ltd. (Nanjing, China), in accordance with the ASTM D2863-2008 standard. The investigated spline size was 130 mm × 6.5 mm × 3 mm. A UL-94 vertical burning tester was used on the CZF-5 (Jiangning, China) according to the ANSI/UL-94-2010 standard. The spline size was 130 mm× 13 mm × 3 mm Calorimetry was carried out on cone-shaped test samples using a calorimeter from FTT Company (London, UK). The sample size was 100 mm × 100 mm × 1.2 mm according to the ISO5660-1 standard. The test power applied was 50 kW·m^−2^, leading to a radiation cone temperature of ≈ 756 °C. The preparation of the LIG coating via laser scanning was carried out on the entire surface of different APP/EP flame-retardant test samples to obtain LIG/APP/EP composites. During the cone tests, the samples were wrapped in double-layer aluminum foil and placed horizontally on the test tray.

According to the intensity ratio of the G peak to the D peak (*I_G_*/*I_D_*) in the Raman spectroscopy data, the *L_a_* value was calculated using Equation (1) [28]:(1)$$L_{a}=2.4\times 10^{-10}\lambda_{l}^{4}\left(\frac{I_{G}}{I_{D}}\right)$$
where *λ_l_* is the wavelength of the Raman laser (*λ_l_* = 532 nm).

### 2.3. Fabrication of a Supercapacitor Electrode and the Electrochemical Test 

The LIG powders were scratched off from the APP/EP substrate; the LIG powders were then mixed and ground uniformly with conductive carbon black and PTFE (polytetrafluoroethylene) in a ratio of 8:1:1. The mixture was then placed in sonicated deionized water and ultrasonically dispersed for 30 min. A certain amount of the uniform mixed solution was dropped onto clean foamed nickel using a pipette gun, which was followed by drying for 12 h in an oven. Electrode samples with a mass of ≈ 1–2 mg were selected for testing.

The electrochemical performance of the LIG supercapacitor working electrode was first evaluated in terms of the galvanostatic charge–discharge (GCD) curves of all the electrode devices, which were recorded via cyclic voltammetry (CV) with a three-electrode configuration using a CHI 660 workstation (Shanghai, China). A Pt plate and Hg/HgO (in 3 M KOH) were applied as the counter and reference electrodes, respectively.

### 2.4. Fabrication of the LIG-APP/EP Sensors

Using control software (Beijing, China), 4 mm × 20 mm rectangle and 13 mm × 13 mm square patterns were laser-scribed onto the surface of the APP/EP, leading to the conversion of the APP/EP surface into a graphene layer. Two copper wires were connected to the LIG layer with silver paint and the samples were heated at 80 °C for 30 min. Lastly, a polydimethylsiloxane (PDMS)/hexane (10 wt.% PDMS) mixture solution was dropped onto the surface of the porous LIG layers. To protect the copper wires from high temperatures, Kapton PI tape (Shenzhen, China) was used to define the area of the copper wires. The sample was then heated to 80 °C with a holding time of 3 h.

### 2.5. Characterization of the LIG-APP/EP Sensor

For the measurement of the temperature-sensing capability, the prepared LIG-APP/EP sensor was placed in an oven and the Cu wires were connected to the resistance tester for real-time monitoring of the resistance of the LIG-APP/EP sensors. Here, the drying oven served as a medium for controlling the testing temperature. The temperature range was 20–120 °C. Moreover, according to the shape memory effect of the APP/EP composite, it was demonstrated that the resistance could be restored to the approximate initial value before the deformation occurred. In this section, for convenience, *R* and *R*_0_ are the resistances of the LIG-APP/EP sensor before and after the temperature changes, respectively. Visual evidence of the temperature-induced deformations is shown in the Appendix A.

## 3. Results

### 3.1. Surface Morphology and the Structure Characterization

Figure 1a demonstrates the processing of the APP/EP composite. The mixture of EP, APP, and m-PDA was injected into the PTFE molds of different sizes to prepare graphene coatings with different structures. The conductive LIG film was prepared using a facile laser-scribing technology on the surface of the APP/EP composite, as illustrated in Figure 1b. The successful preparation of the LIG on APP/EP was attributed to the extreme thermal radiation emitted from the patterning CO_2_ laser (10.6 μm) in air conditions. As shown in Figure 1b, some black LIG layers were generated on the APP/EP surface. The SEM image of the cross-section of the laser-scribed APP/EP (Figure 1c) demonstrated that the thickness of the LIG layer was ≈50–60 μm. The thickness of the LIG layer could be controlled via adjusting the laser parameters [26]. The laser-scribed APP/EP was converted into LIG, whereas the unexposed areas remained unchanged, as depicted in Figure 1d. Because of the fast thermal decomposition of the APP/EP and the release of gas during the laser scribing, the prepared LIG layer showed considerable porosity and consisted of a uniform 2D hierarchical structure (Figure 1e). The high-resolution TEM (HRTEM) image (Figure 1f) shows that the LIG consisted of a series of graphene layers with an abundant wrinkled structure. The number of LIG layers was less than 10. Furthermore, it is illustrated that between the clear graphene stripes, the average lattice space was 0.34 nm, which corresponded to the distance between the adjacent (002) planes in the graphite materials, and it was well matched with the analysis of the XRD pattern (as shown in Figure 1i).

The LIG displayed three outstanding characteristic peaks in the Raman spectrum (Figure 1g), which were clearly different from the pristine APP/EP without laser scribing. The three typical LIG peaks corresponded to the D peak at ≈ 1350 cm^−1^, the G peak at ≈ 1580 cm^−1^, and the 2D peak at ≈ 2700 cm^−1^. The 2D peak located at 2700 cm^−1^ is indicative of typical double-layer graphene-based materials [29,30], proving the formation of several layers of graphene. It can be concluded that the LIG was a highly graphitized carbon that was acquired owing to high temperature (>2500 °C), which is different from the carbon-based materials with a low degree of graphitization or even amorphous carbon [31]. Figure 1h shows the separate XPS spectra of LIG and the APP/EP composite. Compared with the original APP/EP matrix, the XPS spectrum of the LIG showed intense C peaks and faint O, P, and N peaks, which demonstrated that the laser effectively decomposed the APP/EP composite, which involved the removal of oxygen-containing functional groups. The XRD pattern also confirmed the generation of LIG (Figure 1i). A typical peak was observed at 25.9°, corresponding to the (002) plane of the graphitic crystalline phase. The peak at 42.9° corresponded to the (100) reflection, which could be ascribed to an in-plane structure. The combined results of the Raman spectroscopy, XPS, and XRD (Figure 1g–i) clearly evidenced the existence of a graphitic structure.

The Raman curves for the LIG with different APP contents (0–20%) are shown in Appendix A, and the insert images show the I_G_/I_D_ value. As the APP content reached 15%, the degree of graphitization displayed a maximum, whereas, for a 20% APP content, a slight decrease occurred. Therefore, the range of the added amount of APP in the subsequent test was chosen to be 0–15%. The influence of the number of laser passes on generating the LIG was also investigated by tuning the laser-scribing parameters. One set of LIG samples was obtained by applying one to six laser-scribing passes (LIG 1× to LIG 6×) with a laser power of 5 W and a scribing speed of 10 mm·s^−1^. As mentioned above, the degree of graphitization was characterized using Raman spectroscopy. Figure 2a illustrates that the intensity of the G peak reached a maximum with four passes, which may well correspond to the maximum degree of graphitization. The degree of graphitization gradually increased as the scribe passes increased from one to four, which could be attributed to the increased temperature during the multiple laser passes. On the other hand, the degree of graphitization seemed to decrease when the number of scribe passes was increased above four. This may have been due to the oxidation reaction or high-temperature ablation in the ambient atmosphere. The *I_G_*/*I_D_* ratio value obtained from the Raman curves versus the laser-scribing passes provided a similar result, as depicted in Figure 2b. In addition, the L_a_ value along the a-axis (graphene crystalline size) in the LIG obtained via Raman spectroscopy was calculated using Equation (1). The maximum was at 64.3 nm with four laser passes. It was, thus, clearly indicated that four laser passes was the most appropriate choice for preparing LIG. The graphene size reduced with more laser passes, whereas the *L_a_* value decreased to 24 nm as the number of laser passes increased to six. This was further confirmed by the deep defect-correlated D’ peaks around ≈ 1.620 cm^−1^ in the LIG with six laser passes.

### 3.2. Influence of the Flame-Retardant Performance and Laser Process Parameters on the LIG Formed on the APP/EP Composite

The degree of graphitization of the prepared LIG and the number of graphene layers mainly depended on the flame-retardant properties of the substrate and the laser process parameters; the detailed information is shown in Figure 2.

The Raman curves for the LIG with different APP contents (0–20%) are shown in Figure 2a, and the insert images show the *I_G_*/*I_D_* value (the value represents the degree of graphitization). As the APP content reached 15%, the degree of graphitization displayed a maximum, whereas, for an APP content of 20%, a slight decrease occurred. Therefore, the range of the added amount of APP in the subsequent test was chosen to be 0–15%.

Figure 2b and Appendix A illustrate that an increasing concentration of APP led to a decrease in the decomposition temperature of the APP/EP composite materials. This was due to the barrier effect of the APP flame retardant, which decomposed first by producing metaphosphate, which again promoted the formation of carbon layers (Appendix A). The EP displayed a stage of heat degradation, which corresponded to the thermal decomposition of the epoxy chains (400–500 °C) in a nitrogen atmosphere. Compared with the pure EP, the APP/EP composite displayed a lower initial thermal decomposition temperature (*T*_onset_) because of the addition of the APP flame retardant, which indicated the facile and rapid heat absorption of the composites. As the added amount of APP increased to 15%, the decomposition temperature decreased to 327 °C. However, the weight of the residual carbon increased significantly from 16 to 36% at 600 °C. This again indicated that LIG could be generated on the APP/EP composite surface. Furthermore, as the red curves in Figure 2b show, the LIG prepared on the surface of the APP/EP composite remained stable above 700 °C. Thus, the TG curves verified that the LIG had good stability at high temperatures, which means that LIG can play a flame-retardant role.

The flame-retardant performance of pure EP, APP/EP, and LIG/APP/EP were studied via LOI testing, UL-94 vertical burning tests, and cone calorimeter (CC) tests. The LOI results are shown in Figure 2c. The pure EP exhibited a low LOI value of 24% and hardly any self-extinguishing behavior (no level in the UL-94 vertical burning test) (Figure 2c). By incorporating 15 wt.% of APP into the EP, the LOI value increased significantly to 33.4% due to the flame-retardant effect of the APP, whereas the UL-94 reached the V0 level. Further investigations of the flame retardancy of the LIG/APP/EP via LOI and UL-94 testing revealed that the LOI could reach values of up to 37% and the UL-94 remained at the V0 level. Accordingly, we confirmed that when the amount of APP added was 15%, we obtained the optimal value of graphitization degree.

When the amount of added APP was 15%, we illustrated the influence of different laser passes on the degree of graphitization through the Raman spectrum test. Figure 2d illustrates that the intensity of the G peak reached a maximum for four passes, which may well correspond to the maximum degree of graphitization. The degree of graphitization gradually increased as the number of scribe passes increased from one to four, which may be attributed to the increased temperature during the multiple laser passes. On the other hand, the degree of graphitization seemed to decrease when the number of scribe passes was increased above four. This may have been due to the oxidation reaction or the high-temperature ablation in the ambient atmosphere. The ratio value of *I_G_*/*I_D_* obtained from the Raman curves versus the number of laser-scribing passes provided a similar result, as depicted in Figure 2e. In addition, the *L_a_* value along the *a*-axis (graphene crystalline size) in the LIG obtained via Raman spectroscopy was calculated using Equation (1). The maximum was at 64.3 nm with four laser passes (as shown in Figure 2f). It was, thus, clearly indicated that using four laser passes was the most appropriate choice for preparing the LIG. The graphene size reduced with an increased number of laser passes, whereas the *L_a_* value decreased to 24 nm as the number of laser passes increased to six. This was further confirmed by the deep defect-correlated D’ peaks at ≈ 1.620 cm^−1^ in the LIG with six laser passes. 

Notably, the resistance (*R*) of the LIG (1 cm × 1 cm) had a transition with the increasing number of laser passes (Appendix A). The resistance decreased from 2.87 × 10^4^ to 9 Ω as the number of laser passes gradually increased from one to four. On the other hand, an increase in *R* from 9 to 33 Ω was detected as the number of laser passes was increased to six. This may indicate that the energy transferred by the laser had a distinct effect on the transformation of the APP/EP to LIG, which in turn may have led to the observed significant variation in the resistance. This finding implies that an insufficient number of laser passes or a low APP content may lead to a failed LIG transformation. An insufficient number of laser passes resulted in a too low imported energy per unit area, whereas a low APP content led to the high flammability of the epoxy resin.

In summary, the parameters selected for our subsequent tests were as follows: APP addition—15%, laser power—5 W, laser scanning speed—10 mm·s^−1^, and number of laser passes—four.

### 3.3. The Growth Mechanism of the LIG Formed on the APP/EP

Figure 3 shows the laser-induced transformation mechanism of the APP/EP into LIG under ambient air conditions. A photothermal effect may be introduced during direct laser scribing, which leads to a high local temperature, combustion erosion, and depolymerization of the APP/EP composite. Due to the barrier effect of APP [32], APP degradation produces phosphoric acid and metaphosphoric acid that can catalyze the EP into amorphous carbon. Therefore, the surface of APP/EP may be photothermally converted into amorphous carbon first [31], whereas subsequent exposures to the laser scribing may facilitate the conversion of the amorphous carbon into graphene (with four passes being the optimal number). Figure 3 illustrates that the high temperature produced by the laser scribing would destroy the C=O and N−C bonds [33,34,35], which was confirmed by the elemental decrease of oxygen and nitrogen in the LIG. Certain amounts of N, P, and C atoms would most likely recombine and “recrystallize” into N- and P-doped graphene under the release of gaseous products, such as H_2_O, CO_2_, and N*_x_*O*_y_* [36]. The rather rapid emission of gaseous products along the path of the laser scribing may explain the generation of several layers of LIG and 3D porous structures.

### 3.4. The Application of a Fire-Protection Coating Based on the LIG Formed on the APP/EP

The high stability of the prepared LIG (the red line in Figure 2b) remained stable above 700 °C. Thus, we prepared a layer of LIG on the surface of the APP/EP to be used as a flame-protection coating, where the laser process was the same as in Section 3.2. The LOI value and UL-94 vertical test was analyzed using Figure 2c. To further analyze the flame-retardant behavior of the LIG/APP/EP coating, cone calorimeter tests were carried out, the results of which are shown in Figure 4 and Appendix A, and Table 1.

The results of the heat release rate (HRR), total heat release (THR), total smoke production (TSP), and smoke production rate (SPR) tests are shown in Figure 4a–c and Appendix A, and Table 1. In order to validate the accuracy of the results, two tests were performed for each sample. Pure EP exhibited high HRRs (measured twice) of ≈ 1292.6 kW·m^−2^ and 1667.9 kW·m^−2^, and THRs of ≈ 60.2 mJ·m^−2^ and 62.2 mJ·m^−2^. The combustion of the EP led to a considerable amount of smoke, where the TSPs were 16.43 m^2^ and 17.66 m^2^ and the SPRs were 0.27 m^2^·s^−1^ and 0.33 m^2^·s^−1^. By adding APP to the pure EP to form the APP/EP composite, an effective improvement in the flame retardancy was achieved. This was manifested in the values of the peak heat release rates (PHRRs) (812.1 kW·m^−2^ and 955.9 kW·m^−2^), THRs (28.1 MJ·m^−2^ and 24.5 MJ·m^−2^), TSPs (5.65 m^2^ and 5.45 m^2^), and SPRs (0.18 m^2^·s^−1^ and 0.17 m^2^·s^−1^), which corresponded to reductions of ≈ 37.2%, ≈ 53.3%, ≈ 65.6%, and ≈33.3%, respectively. After the laser scribing and adding an LIG layer to the APP/EP composite, a further improvement of the flame retardancy was observed, which is shown in Figure 4a–c and Table 1. Values of the PHRRs (374.1 kW·m^−2^ and 286.2 kW·m^−2^), THRs (14.9 MJ·m^−2^ and 19.9 MJ·m^−2^), TSP (4.26 m^2^ and 4.85 m^2^), and SPR (0.16 m^2^·s^−1^ and 0.10 m^2^·s^−1^) were determined, which corresponded to reductions of ≈ 71.1%, ≈ 75.2%, ≈ 74.1%, and ≈ 40.7% as compared to pure EP, respectively. The values for the time to ignition (TTI) exhibited a dramatic increase compared to pure EP from 21 s and 23 s to 57 s and 53 s, which would increase the burning time by up to ≈ 34 s in a real fire hazard. Based on these results, combined with the significant improvement in the LOI value (37%) and the V0 level of the UL-94 test, it can be concluded that the fabrication of the LIG coating layer on the APP/EP composite led to notable synergistic enhancements in the flame-retardant performance and smoke-suppressant property. Compared with traditional methods for improving flame retardancy, the above results constituted a significant enhancement in the flame-retardant behavior [37].

### 3.5. The Application of a Supercapacitor Electrode Based on the LIG Formed on the APP/EP

Through the XPS test (as shown in Figure 1), the prepared LIG contained four elements, namely, C, N, O, and P, which shows that the LIG was doped with N and P. To further verify this conclusion, we carried out peak analysis based on the XPS test, as well as an EDX test. Applying the prepared LIG as the electrode material of the supercapacitors verified the excellent electrochemical performance of the LIG.

The element identification and the states of the carbon, oxygen, nitrogen, and phosphorus were studied. The high-resolution images of the C 1s spectrum in Figure 5a displayed four prominent peaks: C–C (≈284.6 eV), C–O (≈285.5 eV), O=C–N (≈286.7 eV), and O–C=O (≈289.0 eV) [38]. As shown in Figure 1h, we note that there was a large amount of nitrogen and phosphorus in the LIG films. The high-resolution spectrum of N 1s in Figure 5b shows evidence of four typical N peaks, which were assigned to N-6, N-5, N-Q, and N-O, which were from pyridinic (≈398.5 eV), pyrrolic (≈399.9 eV), graphitic (≈401.2 eV) and oxidized N (≈403.8 eV) [39]. The chemical reaction of the decomposition of APP/EP during the laser scribing was rather complicated, involving the generation of several reaction intermediates and products. Therefore, it may be plausible that N atoms with different valence states can be detected in the LIG. It is rather energy and time consuming to prepare graphitic nitrogen through a hydrothermal method according to previous reports [40], while laser scribing involving several passes could rapidly and efficiently lead to nitrogen-doped porous graphene. Figure 5c also shows the high-resolution image of the P 2p XPS spectrum of the LIG, which was collected to gain more detailed information about the phosphorus doping. As illustrated in Figure 5c, two prominent peaks located at 130.1 eV and 132.0 eV could be assigned to the P–C and P–O bonds, respectively [41]. This result indicates that the phosphorus atoms were successfully doped into the LIG through laser scribing. Accordingly, the obtained LIG material was a porous graphene material that was doped with both N and P atoms.

The EDX maps indicated a uniform dispersion of C, N, P, and O elements in the LIG. The mass loading obtained from the EDX analysis suggested a 6.38% N and 7.69% P coverage on the LIG that was formed on the surface of the APP/EP composite. From these results, combined with the XPS data (see Figure 5d–h), it can be confirmed that the LIG was indeed made of graphene material that was co-doped with N and P. It was reported previously that the electrochemical performance of N- or P-doped graphene is significantly better than undoped graphene [38,41,42].

The LIG formed on the surface of the APP/EP by four passes of laser scribing was used as the electrode material in a supercapacitor. The CV curves for the LIG in a 3 M electrolyte (KOH) at different current densities were tested using a three-electrode system (Figure 5i). As illustrated in Figure 5i, the LIG electrodes that were fabricated using various numbers of laser passes exhibited an approximately rectangular shape, which corresponds to the ideal behavior of electric double-layer capacitors. At a scanning rate of 1 V·s^−1^, no pseudocapacitive performance was indicated and no anodic and cathodic peaks are shown in Figure 5i. Even when the scan rate rose to 10 V·s^−1^, the CV curves maintained their pseudorectangular shape, which is indicative of high-power performance [43]. This is clear evidence that LIG is a potential electrode material. This excellent capacitive performance was confirmed via GCD curves at different current densities ranging from 0.025 to 0.6 A·g^−1^ (Figure 5j). The specific capacitance (Csc) was calculated from the GCD curves (Figure 5k), where the highest specific capacitance obtained was ≈245.6 F·g^−1^. This value was higher than that for the LIG with undoped hybrid atoms reported in previous studies [24], and it was also higher than the values acquired in previous studies on GO-derived supercapacitors [44,45,46,47]. The high specific capacitance value of the LIG supercapacitor encountered here may be ascribed to the hierarchical porous structure of the highly conductive LIG coating on the EP surface. This structural feature had a high surface area with plentiful wrinkles and provided facile access for the formation of Helmholtz layers at the contact between the electrolyte and the LIG-APP/EP electrode. It is anticipated that these findings validate this innovative approach here and may inspire scholars to further investigate the mechanisms and explore the potential of other polymers.

### 3.6. Application of the Temperature Sensor Based on the LIG Formed on the APP/EP 

According to the above results and inspired by traditional carbon-material-based sensors, we prepared several thin-film resistance-type sensors on the surface of the APP/EP composites via laser scribing. Figure 6a illustrates the structure of the LIG-APP/EP sensors, which could respond to the temperature changes of the LIG-APP/EP. The LIG-APP/EP sensors were fabricated on top of the APP/EP matrix, followed by the LIG layers, a PDMS film, PI tape, and copper foil electrodes. In addition, silver (Ag) paste was applied to connect the LIG layers and copper foil electrodes. From the SEM image of the cross-section of the LIG-APP/EP sensors in Figure 6b, it was concluded that the thickness of the LIG conductive layer was ≈ 60 μm. Furthermore, due to the porous structure of the LIG layer, it can be safely assumed that the low concentration liquid PDMS precursor (10 wt.%) would easily penetrate the LIG layer and firmly combine with the APP/EP substrate. In addition, PDMS is resistant to high temperatures, which implies that it may be used for prolonged times at the maximum operating temperature of the LIG-APP/EP sensor of 200 °C.

As discussed above, the LIG layer on the APP/EP surface displayed good electrochemical performance and high conductivity (the effective conductive layer was a 4 mm× 20 mm rectangle), combined with good flame retardancy. The resistance changing with the temperature allows for application of the device as a temperature sensor. As illustrated in Figure 6c and Appendix A, the resistance–temperature curves displayed a positive temperature coefficient (PTC) of resistance in the temperature range of 20–120 °C [48]. Therefore, possible temperature changes in an electronic device can be monitored via a change in the resistance of the LIG-APP/EP sensor to avoid accidents caused by excessive temperature. In addition, the schematic diagram of the resistance changes is shown in Figure 6d, where such changes would be caused by high-temperature bending deformation of the LIG-APP/EP sensor and the shape memory effect of the EP matrix. When the LIG-APP/EP sensor was heated above the glass transition temperature, the bending deformation reached a maximum bending value, which remained fixed. Even by cooling to room temperature, the LIG-APP/EP sensor maintained the deformed shape (as shown in Appendix A). The resistance of the LIG-APP/EP sensor changed from the initial *R*_0_ ≈ 37 Ω to no resistance after the sample had been cooled down to room temperature in the bent state. As the sensor bent outward, the LIG film sheets were isolated from each other and formed cracks; thus, the LIG thin layer did not conduct electricity and the resistance did not exist. At room temperature, the shape remained unchanged. By placing the deformed LIG-APP/EP sensor into a high-temperature drying oven at ≈ 180 °C, the LIG-APP/EP sensor gradually returned to its original shape and the resistance gradually decreased (R_1_ ≈ 106 Ω) because the LIG film sheets were isolated from each other. When it reached the state just before deformation, the resistance was close to the *R*_0_ value (≈35 Ω). This result may be explained by the internal cross-linking of the APP/EP composite, which caused reversible softening and hardening changes. Therefore, the LIG-APP/EP sensor could also monitor the resistance changes during the deformation. Figure 6e displays the application of the LIG-APP/EP sensor in hot water. As mentioned above in the Section 2.4, the effective conductive layer was a 13 mm × 13 mm square. The resistance of the original LIG-APP/EP sensor was 21.66 Ω at 15 °C, which increased to 30.02 Ω when the temperature increased to 80 °C.

The excellent temperature-sensing performance of the LIG layer fabricated on the APP/EP surface may be attributed to the specific structure of the conductive LIG coating and the shape memory effect of the APP/EP composite. The porous hierarchical LIG sheets could respond to the temperature changes in the device. Furthermore, the laser-scribing process allowed for tailoring the sensors to many different modes. This strategy of integrating multiple sensors into one component allowed for monitoring the changes in temperature and deformation. The LIG-APP/EP sensors were also fully functional in a hot water environment, which indicates that the LIG sensor may be applied in different environments, i.e., in air and hot water. When the deformed APP/EP is used in a practical application, it can also monitor the temperature changes of electronic components through the resistance change. It was confirmed beyond any doubt that the laser-scribing method on EP composites has significant potential applications in various smart electronic components.

## 4. Conclusions

In this study, an intelligent fire-protection coating was fabricated on the surface of an APP/EP composite using laser scribing under ambient air conditions. Highly flame-retardant behavior and potential applications as temperature sensors in various environments were demonstrated. Direct laser scribing and multiple laser passes on the APP/EP led to a photothermal effect and a high local temperature, which resulted in the depolymerization of the APP/EP surface and the generation of graphene layers. The LIG formed on the APP/EP had a porous hierarchical graphene surface structure. The LIG was fabricated by applying a laser power of 5 W and a scribing speed of 10 mm·s^−1^. It was shown that four scribing passes led to an optimal graphene structure and application performance. The prepared thin graphene coatings (LIG) increased the TTI by about 36 s, and the PHRR, THR, and TSP values all showed significant decreases of ≈ 71.1%, ≈ 75.2%, ≈ 74.1%, and ≈ 40.7%, respectively, which demonstrated a massively improved flame retardancy. Furthermore, we prepared a high electrochemical performance supercapacitor with LIG electrodes, which added further functionality to the device. Furthermore, temperature sensors were fabricated directly onto the LIG surface, displaying a metallic-type PTC effect. The temperature sensors were applied in a hot water environment, showing that they could monitor the temperature changes of electronic components in different conditions. We foresee that the laser-scribing technology that was used for forming graphene on the APP/EP surfaces may be applied to other polymers, which may lead to a quick and efficient improvement in their flame-retardant performance and electrical properties.

## Figures and Tables

**Figure 1 polymers-13-00984-f001:**
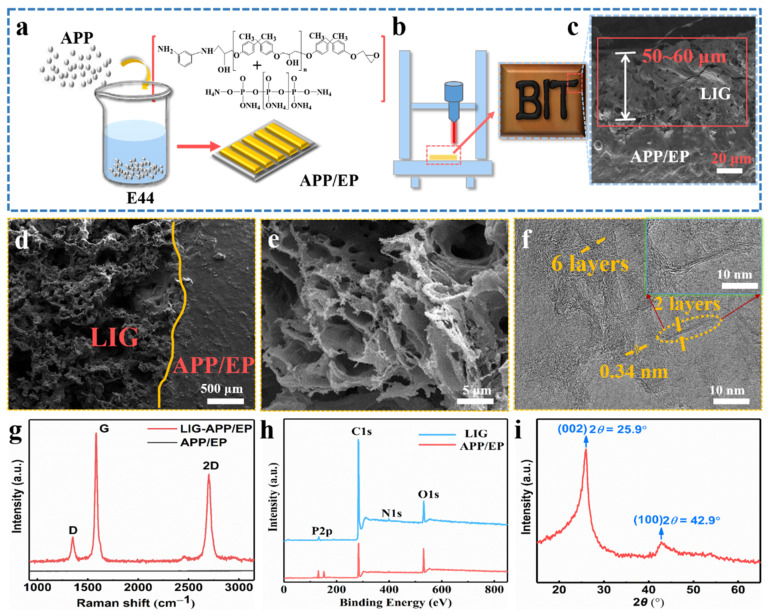
Fabrication of the specimens and the characterization of the laser-induced graphene (LIG). (**a**) Processing route of the ammonium polyphosphate and epoxy (APP/EP) composites. (**b**) Processing route for fabricating an LIG layer on the APP/EP surface. The inset shows a photograph of the LIG patterned into “BIT” on the APP/EP. (**c**) Scanning electron microscopy (SEM) image of the cross-section of the LIG sample. (**d**) Enlarged SEM cross-section image of the LIG area. The smooth region was the APP/EP substrate. The porous region on the left was the LIG layer. (**e**) High-resolution SEM image of the LIG formed on the APP/EP. (**f**) High-resolution transmission electron microscopy (HRTEM) image of the LIG. The yellow arrows mark the few layers of graphene. The average lattice spacing was 0.34 nm. (**g**) Representative Raman spectrum of the LIG layer and the APP/EP composite, where the illustration is a magnified view of a location marked in yellow in (**f**). (**h**) Representative X-ray photoelectron spectroscopy (XPS) spectrum of the LIG layer and the APP/EP composite. (**i**) X-ray diffraction (XRD) spectrum of the LIG scraped from the APP/EP. The (002) planes of the graphitic materials corresponded to the 0.34 nm distance marked in the HRTEM images.

**Figure 2 polymers-13-00984-f002:**
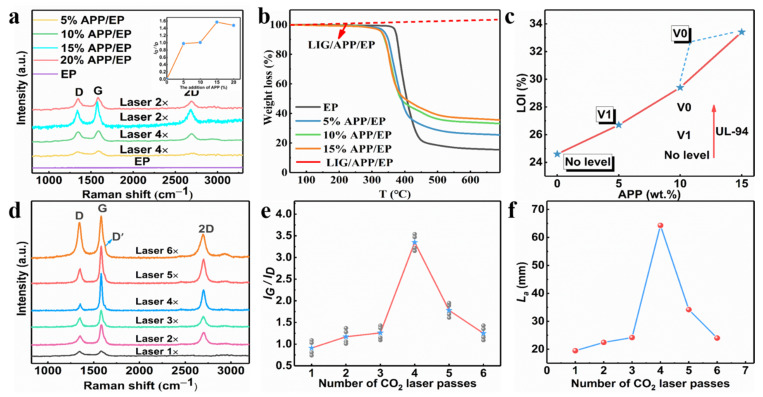
(**a**) TGA curves for the APP/EP composites with different APP content and LIG/APP/EP in an air atmosphere. (**b**) The limiting oxygen index (LOI) value (%) and the UL-94 rating for the EP and APP/EP composites with different APP contents. (**c**) Raman spectrum curves of the LIG formed on APP/EP with different APP contents and the numbers of laser passes, and the inset is the corresponding *I_G_*/*I_D_* ratio. (**d**) Raman spectra of the LIG generated with different numbers of laser passes. (**e**) The integrated intensities of the G and D peaks (*I_G_*/*I_D_*) for different LIGs. (**f**) The crystalline size (*L_a_*) along the *a*-axis of different LIGs.

**Figure 3 polymers-13-00984-f003:**
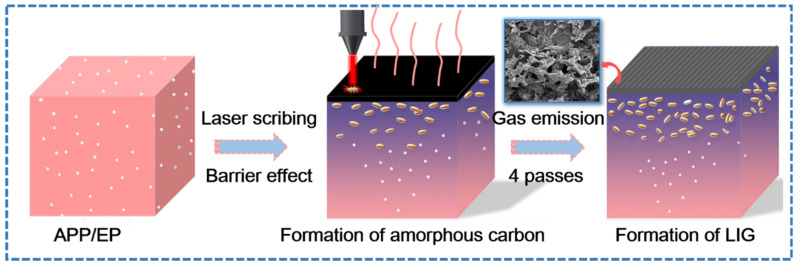
The growth mechanism of the LIG formed on the APP/EP.

**Figure 4 polymers-13-00984-f004:**
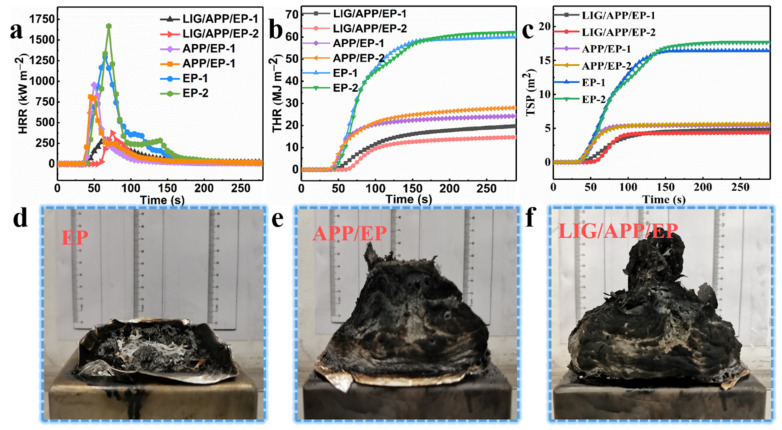
The results of the cone test. (**a**) Heat release rate (HRR), (**b**) total heat release (THR), and (**c**) total smoke production (TSP) curves of the EP, APP/EP, and LIG/APP/EP. Digital photo images of the residual carbon from the cone tests of the (**d**) EP, (**e**) APP/EP, and (**f**) LIG/APP/EP.

**Figure 5 polymers-13-00984-f005:**
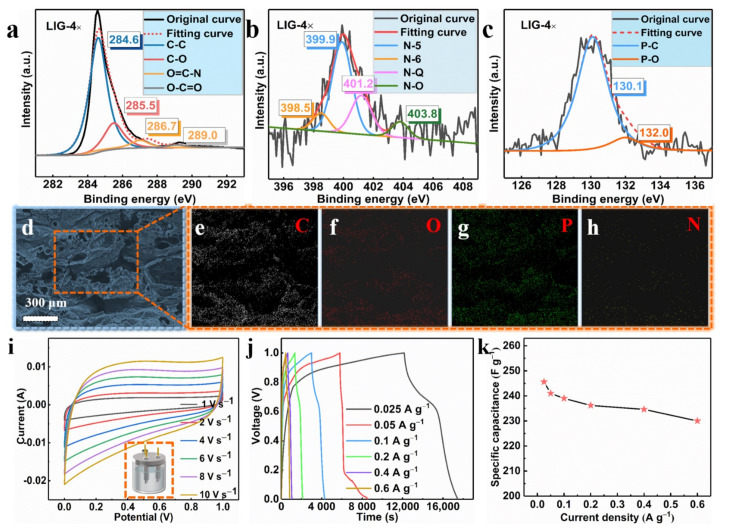
(**a**) C 1s XPS spectrum of the LIG obtained using laser scribing (LIG 4×). (**b**) N 1s XPS spectrum of the LIG generated by four laser passes (LIG 4×). (**c**) P 2p XPS spectrum of the LIG prepared using four laser passes (LIG 4×). (**d**) SEM image of the LIG formed on the surface of APP/EP. The corresponding EDX results for C (**e**), O (**f**), P (**g**), and N (**h**). (**i**) Cyclic voltammetry (CV) curves of the LIG/EP electrode at scan rates ranging from 1 to 10 V·s^−1^. (**j**) Galvanostatic charge–discharge (GCD) curves of the LIG-EP electrode at various current densities ranging from 0.025 to 0.6 A·g^−1^. (**k**) The specific capacitance (Csc) of the LIG/EP in the current density range from 0.025 to 0.6 A·g^−1^.

**Figure 6 polymers-13-00984-f006:**
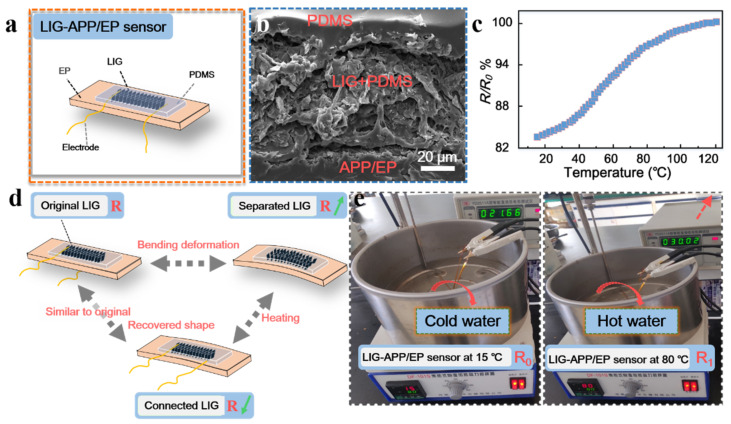
Performance and structure of the LIG sensors. (**a**) Structure of the LIG sensors. LIG and APP/EP were packaged together with a solvent based on polydimethylsiloxane (PDMS). (**b**) SEM image illustrating the cross-section of the LIG sensors. (**c**) The resistance–temperature curve between 20–120 °C. (**d**) Working mechanism of the LIG-APP/EP sensors for temperature detection based on the shape memory of the EP composite. (**e**) Illustration of the water-resistant LIG-APP/EP sensor for detecting changes in temperature.

**Table 1 polymers-13-00984-t001:** Combustion data of the neat EP, the APP/EP, and the LIG/APP/EP.

Sample	TTI (s)	PHRR (kW·m^−2^)	THR (MJ·m^−2^)	TSP (m^2^)	SPR (m^2^·s^−1^)
LIG/APP/EP	57	374.1	14.9	4.26	0.16
LIG/APP/EP	53	286.2	19.9	4.85	0.10
APP/EP	25	812.1	28.1	5.65	0.18
APP/EP	27	955.9	24.5	5.45	0.17
EP	21	1292.6	60.2	16.43	0.27
EP	23	1667.9	62.2	17.66	0.33

TTI: time to ignition, PHRR: peak heat release rate, SPR: smoke production rate.

## Data Availability

The data presented in this study are available on request from the corresponding author.

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
