# Peer review of "An Intelligent Fire-Protection Coating Based on Ammonium Polyphosphate/Epoxy Composites and Laser-Induced Graphene"

_polymers, 2021, doi:10.3390/polym13060984_

Round 1
Reviewer 1 Report
Due to its outstanding properties graphene is of increasing interest for several applications, for instance as part of flame-retardant systems for polymers. The subject of the presented study is the laser-induced formation of graphene layers at the surface of an epoxy material containing ammonium polyphosphate (APP). APP is widely used as flame retardant in polymers because it causes the formation of protective carbonized layer when the polymer is exposed to heat. The authors could provide evidence that the photo-thermal effect of laser beam not only induces carbonization, but may transform the carbonized layer into graphene (laser-scribing). Investigations of the flame-retardant properties by means of UL 94 test, LOI test and cone calorimetry revealed additional flame retardant effect of thin graphene layers obtained by laser-scribing. Most importantly, the time to ignition of the epoxy resin samples were found to be considerably increased and the smoke release was diminished.
Moreover, the authors investigated the usability of the laser-induced graphene for additional applications (supercapacitor electrodes, temperature sensor). These investigations yielded promising results.
Due to its interesting content I recommend the manuscript of Liu et al. for publication in the journal polymers after removing of few grammar and spelling errors (for instance, please replace the word “where” by “were” at page 9, line 351, please remove the comma after the word “matrix” at page 13, line 449).
Author Response
Reviewer 1:
Due to its outstanding properties graphene is of increasing interest for several applications, for instance as part of flame-retardant systems for polymers. The subject of the presented study is the laser-induced formation of graphene layers at the surface of an epoxy material containing ammonium polyphosphate (APP). APP is widely used as flame retardant in polymers because it causes the formation of protective carbonized layer when the polymer is exposed to heat. The authors could provide evidence that the photo-thermal effect of laser beam not only induces carbonization, but may transform the carbonized layer into graphene (laser-scribing). Investigations of the flame-retardant properties by means of UL 94 test, LOI test and cone calorimetry revealed additional flame retardant effect of thin graphene layers obtained by laser-scribing. Most importantly, the time to ignition of the epoxy resin samples were found to be considerably increased and the smoke release was diminished.
Moreover, the authors investigated the usability of the laser-induced graphene for additional applications (supercapacitor electrodes, temperature sensor). These investigations yielded promising results.
Due to its interesting content I recommend the manuscript of Liu et al. for publication in the journal polymers after removing of few grammar and spelling errors (for instance, please replace the word “where” by “were” at page 9, line 351, please remove the comma after the word “matrix” at page 13, line 449).
Reply:
Thank you for your high evaluation and suggestions on our manuscript, which have helped us to further improve our manuscript. And we have modified our manuscript according to your kind suggestions, the changes are marked in BLUE in the revised manuscript.
According to your suggestion, we have replace the word “where” by “were” at page 9, line 35 in BLUE, and deleted the “,” after the word “matrix” at page 13, line 449.

Reviewer 2 Report
The subject of the manuscript focused on an intelligent fire protection coating based on ammonium 2 polyphosphate/epoxy composites and laser induced graphene is in good relevance with the scope of POLYMERS. The manuscript is well written and the content is properly arranged.
However, the text requires the following corrections and additions:
Figure 1 f should be shown in a larger scale because in the present presentation the layers of graphene are not visible and can only be observed after a much greater magnification of this HRTEM image of LIG. The same remark applies to Figures 5 c-h. Moreover, the discussion of the results based on figures 5 d-h is too general. in both cases, the way the results are shown makes their interpretation difficult and does not fully justify the conclusions drawn.
Author Response
Reviewer 2:
The subject of the manuscript focused on an intelligent fire protection coating based on ammonium 2 polyphosphate/epoxy composites and laser induced graphene is in good relevance with the scope of polymers. The manuscript is well written and the content is properly arranged.
However, the text requires the following corrections and additions:
Figure 1 f should be shown in a larger scale because in the present presentation the layers of graphene are not visible and can only be observed after a much greater magnification of this HRTEM image of LIG. The same remark applies to Figures 5 c-h. Moreover, the discussion of the results based on figures 5 d-h is too general. in both cases, the way the results are shown makes their interpretation difficult and does not fully justify the conclusions drawn.
Reply:
Thank you for your good questions and suggestions recommends on our manuscript, which have helped us to further improve our manuscript. And we have modified our manuscript according to your kind suggestions, the changes are marked in BLUE in the revised manuscript.
- Figure 1 f should be shown in a larger scale because in the present presentation the layers of graphene are not visible and can only be observed after a much greater magnification of this HRTEM image of LIG.
Reply: Thanks for your suggestion. I have added an enlarged view of the local location in Figure 1f and explained it in the legend in BLUE, so that the graphene sheets can be observed more clearly.
- The same remark applies to Figures 5 c-h. Moreover, the discussion of the results based on figures 5 d-h is too general. In both cases, the way the results are shown makes their interpretation difficult and does not fully justify the conclusions drawn.
Reply:
Thanks for your good question. Since the LIG-APP/EP prepared in this work is graphene co-doped with N and P, in Figure 1h, we have explained the four components (C, O, N, P)of the prepared LIG-APP/EP, and it is speculated that the prepared graphene contains N, P, to further verify this conclusion, we processed the XPS's C, N, and P element peaks to better judge the existence of N and P, and combined with the EDX analysis test of LIG-APP/EP, the figure 5d-h shows uniform N, P distribution, it is finally determined that the prepared LIG-APP/EP has a N, P co-doped graphene structure. Therefore, in this part of the EDX characterization combined with Figure 1h and Figure 5a-c can be confirmed N, P had been doped.

Reviewer 3 Report
Dear authors,
See my comments

Author Response
Reviewer 3:
The article in Titled “An Intelligent Fire Protection Coating Based on Ammonium Polyphosphate/Epoxy Composites and Laser Induced Graphene “by authors Weiwei Yang et al. has been submitted to Polymers, MDPI, and has been reviewed for its scope, quality of writing, novelty and acceptance to the reads. The article has demonstrated the fire protection coating with self-monitoring ability. The authors presented smart clothing approach in a good dimension. I recommend the article to be considered in Polymers after the following very mandatory revisions.
Reply:
Thank you for your good questions and suggestions recommends on our manuscript, which have helped us to further improve our manuscript. And we have modified our manuscript according to your kind suggestions, the changes are marked in BLUE in the revised manuscript.
- In the abstract line “the human” does not make sense avoid “the”
Reply:
Thanks for your kind suggestion. I have delete the word “the” in the manuscript.
- Same line (line 11) property injury seems to be not technical
Reply:
Thanks for your good suggestion. I have replaced the “injury” with “damage”, and marked in BLUE in the manuscript.
- Abstract again: line 15 (seems a conclusion word) not appropriate place
Reply:
Thanks for your good suggestion. I have put this sentence in the right place, marked in BLUE in the manuscript.
- Line 16 and 17, I cannot see the proof weather the bonds are broken/the statement is not supported by any evidence or misplaced
Reply:
Thanks for your good question. It is verified by related literature that the laser induction technology will break the chemical bond when high temperature is generated (Reference[24]: Jian L, Peng Z, Liu Y, et al. Laser-induced porous graphene films from commercial polymers[J]. Nature Communications, 2014, 5, 5714.), in this work, the XPS spectrum of LIG-APP/EP is subjected to element peak separation processing. From Figure 5 ac, the bond states of the four elements C, O, N, and P can be observed.
- What is the expected hydrogen to carbon ration in the said polymer?
Reply:
Thanks for your good question. I guess you mean to the carbon-oxygen ratio of the prepared LIG-APP/EP. In this work, the carbon-oxygen ratio tested by XPS is 7 : 2.
- I cannot see any result on SM effect while you put a shape memory in your key word
Reply:
Thanks for your good question. I have added the application of shape memory effect in the summary section, marked in BLUE in the manuscript and verified the shape memory effect in the material through Figure S6. The detailed analysis in page 12 marked in BLUE.
- Better to see the pictorial image of the proof-of-concept
Reply:
Thanks for your kind suggestion, I have added the graphical abstract which is shown below.
Figure 1 The graphical abstract
- The experiment was performed at ambient conditions, better if we can see at varied temperature to see the stability specially the laser scribing
Reply:
Thanks for your kind suggestion, the LIG-APP/EP sensor was applied in the oven with 180 ℃, and which shows good stability. The detailed analysis was shown in page 12, and marked in BLUE.
- Is it possible to see the possible patterns of the graphene?
Reply:
Thanks for your good question. In Figure 1b and Figure S6, we can concluded the different patterns can be obtained by laser technology.
- The last conclusion (line 504-506) should be avoided unless evidence based
Reply:
Thanks for your kind suggestion, the evidence was shown in Figure 6e (page 12). Which was marked in BLUE.
- Is it possible to see the thermal stability? Like TGA and DSC for the composite
Reply:
Thanks for your question. The thermal stability was performed with TGA analysis. And the TGA and DTGA were shown in Figure 2b and Figure S1.
- If you claim the supercapacitor properties, please conduct the experimental tests based on charge and discharge operations with fluctuating DC-current are done and other relevant tests
Reply:
Thanks for your kind suggestion, the prepared LIG-APP/EP was used to fabricate a supercapacitor electrode, and related electrochemical performance tests was shown in Figure 5 i-k, cyclic voltammetry (CV) curves at different scan rates and charge-discharge curves (GCD) at different current densities were measured. The mass specific capacitance of the electrode was calculated according to the charge-discharge test curve.
